# Beyond Gender Binarism: Implications of Sex-Gender Diversity for Health Equity

**DOI:** 10.3390/healthcare13192440

**Published:** 2025-09-26

**Authors:** Peter de-Jesús Villa

**Affiliations:** Research Group GENSEX, Facultad de Ciencias de la Salud, Universidad Fernando Pessoa Canarias (UFPC), 35450 Guía, Spain; pdejesus@ufpcanarias.es

**Keywords:** gender identity, transgender persons, intersex persons, medical education, inclusive research, health equity

## Abstract

The persistence of a binary biomedical framework in healthcare has become increasingly inadequate to address the realities of human diversity. Recent literature highlights how this dichotomous model reinforces inequities for transgender and intersex populations, sustaining barriers to access, stigmatisation, and poorer health outcomes. In this Perspective, I critically reflect on the limitations of the binary paradigm and draw on developments in science, clinical practice, education, and policy to propose a future-oriented approach to health equity. Emerging evidence underscores the complexity of sexual development as a spectrum and the urgent need to move from pathological frameworks toward affirming care based on rights. Key advances include the adoption of affirmative care models, reforms in medical curriculum, and the rise of inclusive research methodologies that capture gender diversity beyond binaries. However, structural barriers—such as rigid clinical protocols, outdated educational content, and insufficient policy alignment—continue to hinder meaningful change. This article advocates for systemic transformation in healthcare education, practice, and research. I outline strategic priorities for the field are the implementation of gender diversity in medical training, the implementation of rights-based clinical guidelines, and the design of inclusive methodologies that remove structural discrimination. These actions are essential to build a more precise, ethical and universally inclusive health system. Ultimately, ensuring sustainable and equitable outcomes requires bridging scientific innovation with human rights principles and focussing on the lived experiences of transgender and intersex individuals.

## 1. Introduction

Our understanding of sex and gender has evolved dramatically over time, shifting from a binary and simplistic view to the recognition of their complexity, multiplicity, and spectral nature.

### 1.1. The Sex/Gender Construct and Its Health and Social Implications

Despite being distinct concepts, sex and gender, are often used interchangeably in social, educational, and healthcare contexts [1,2]. Sex refers to a set of biological characteristics, including chromosomes, gonads, hormones, genitalia, and secondary sexual traits, that influence but do not fully determine an individual’s development [3,4]. Gender, in contrast, is a socially and culturally constructed system encompassing norms, roles, behaviors, and identities associated with being male, female, or other diverse identities [4,5]. The conflation of these concepts has reinforced a dichotomous view (male/female), dominant in Western culture, which assumes that sex assigned at birth, based on a single visible variable—external genitalia—automatically predicts a person’s gender and that both categories are stable and immutable throughout life. Historical and cross-cultural studies, however, show that sex and gender have been understood in more complex, multidimensional ways, and that many societies recognize non-binary and fluid identities [6]. The persistence of a rigid binary model has led to detrimental consequences for transgender, non-binary, and intersex individuals, systematically rendering them invisible or excluded from research, services, and public policies [7,8] highlighting the necessity of integrating biological, social, and cultural dimensions to achieve equitable healthcare and social inclusion.

For example, recent analysis of 1252 prehistoric burials in Central Europe reveals that although the binary model explains 90% of cases with complete data, only about 30% of the graves allowed determination of both sex and gender. A recurring minority deviated from binary norms, suggesting these societies recognized and accepted certain forms of gender diversity in their funerary rituals [9]. This aligns with historically documented non-binary identities across cultures, such as the muxe of Oaxaca [10], hijras in South Asia [11], or the two-spirit identities in Indigenous North American communities [12], Furthermore, it is important to acknowledge that rigid gender binaries have been reinforced through Western colonialism, which imposed binary norms on Indigenous and non-Western societies, often erasing pre-existing understandings of gender diversity [6,13]. Taken together, these findings underscore that strict binarism has neither been universal nor enduring, and that contemporary science and historical evidence both challenge its validity.

Fausto-Sterling [1,14] warns that the binary conception of sex oversimplifies human development, which involves multiple interdependent dimensions: chromosomal, gonadal, hormonal, genital, cerebral, and psychosocial. These layers can vary independently throughout life, producing bodily and experiential diversity that does not fit the classic categories of “man” or “woman” [1,15,16]. Some researchers, propose the term “sex/gender” to reflect the complex interplay between biological characteristics and socially constructed gender roles, advocating for a dynamic biopsychosocial model where biological and environmental influences are inseparable [3,14].

Understanding these complexities is not only theoretically relevant but also essential for healthcare systems. Rigid binary classification based solely on genital obscure the realities of transgender, intersex, and non-binary individuals, limiting access to essential services [17,18], and producing imprecise epidemiological data that compromise the validity of public health models [19]. From a health promotion perspective, adherence to binary norms hinders the development of culturally competent, diversity-sensitive interventions. Preventive or educational programs that assume only two legitimate ways of being and living in a body excludea significant portion of the population—up to 7–8% according to recent estimates [20]—and risk reinforcing structural stigma, adversely affecting the mental, emotional, and physical health of LGBTIQA+ individuals. Consequently, redefining sex and gender is not merely a theoretical or terminological exercise but a public health and ethical imperative. Effective, rights-based healthcare requires transforming educational, legislative, and organizational structures that have historically excluded diverse populations. Equitable, community-centered health promotion must integrate the full spectrum of sex and gender from the outset, grounded in a pluralistic understanding of human bodies and identities [1].

### 1.2. Sexual Development as a Spectrum

Contemporary science demostrates that the traditional concept of male/female binary obscures a more complex reality, and that biological arguments used to deny gender diversity rely on outdated and refuted knowledge [21]. Textbook explanations often reduce sexual development to chromosomal formulas (XX = female, XY = male), from which other characteristics, such as reproductive systems or secondary sexual traits, are inferred. However, scientific advances showthat chromosomes alone do not always determine sex or gender identity. This notion is not new; Fausto-Sterling’s [1,16,22] documented at least five biological sex characteristics -chromosomal, gonadal, hormonal, genital, and gender identity- that do not consistently align with binary categories, underscoring the limitations of the simple male/female dichotomy [1].

Sexual development begins at fertilization with chromosomal sex, yet variations such as XO, XXY, or XYY demonstrate that more than two categories exist even at this earliest stage. By the eighth to twelfth weeks of gestation, fetal gonadal sex develops, guiding hormonal sex, which in turn shapes external genitalia by the fourth month. Puberty adds further complexity as sex hormones drive secondary sexual characteristics [23] such as breast growth or facial hair development.

However, epigenetics may change our understanding of sexual development by introducing gene regulation mechanisms that are dynamic and environment-sensitive [23,24]. For many years, female development was believed to be a “default” state, and male development activated by the SRY gene on the Y chromosome [25]. This idea was disproved by the discovery of genes that actively promote ovarian development [26]. For example, XY individuals with additional copies of the WNT4 gene may present atypical genitalia and gonads, along with rudimentary uterus and fallopian tubes [27]. Likewise, if the RSPO1 gene, critical for ovarian development, malfunctions, XX individuals may develop an ovotestis, a gonad with both ovarian and testicular areas [28]. These findings reveal that gonadal identity arises from a balance between two opposing genetic activity networks, whose equilibrium can be disrupted independently of chromosomes. Some studies suggest this balance may even change postnatally (e.g., [29,30]). Other genes, such as NR5A1, also determine a wide range of effects [31]. Unlike genetic mutations that alter DNA sequence, epigenetic mechanisms affect how genes are turned on or off without changing the underlying DNA sequence. This may have profound implications for sexual development and its variations. Research in this field promises new perspectives with important health repercussions [24,32].

Biology conceives sex as a spectrum where individuals fall at the boundaries between what is considered male or female development, and whose sex chromosomes do not always align with expected gonads or anatomy [19]. Fausto-Sterling [14] notes that at birth, a person has passed through multiple phases of sexual development, and, as with chromosomal sex, each phase does not always conform to a strict binary scheme. Furthermore, each of these layers can vary independently and may not perfectly align with the others, resulting in situations where an XX baby is born with a penis, an XY baby with a vagina, or individuals with XY chromosomes and testes but also a vagina and clitoris who develop breasts during puberty without menstruating, among other possible combinations. That is, some individuals occupy the boundary between what is considered male or female development, with sex chromosomes that do not match expected gonads (ovaries or testes) or sexual anatomy. These individuals are referred to as intersex or as having Differences of Sexual Development (DSD) [19,23], and some studies estimate that up to 1 in 100 people present some form of these conditions [15]. Such variations may be visible at birth (e.g., genital ambiguity), manifest during puberty, or be detected incidentally at any point in life (e.g., [33,34], complicating exact estimation.

This nuanced understanding of sexual development lays the groundwork for reconsidering healthcare practices. Historically, the pressure to conform toa binary sex/gender system has led to coercive medical interventions for individuals whose bodies do not fit dichotomous categories, such as “normalizing” surgeries on intersex newborns [27,28]. The proceduresare often clinically unjustified and may contradict the individual’s later gender identity [14,35,36]. These practices have increasingly been questioned and have led to laws prohibiting irreversible interventions without informed consent, exemplified by Spain’s Law 2/2021 [37]. The American Psychological Association [38] emphasizes that legitimizing such practices—including non-consensual corrective surgeries, misuse of language, or systematic invalidation of identities—constitutes malpractice and volates core ethical principles of respect, justice, and non-discrimination.

## 2. Gender Identity and Social Determinants of Health

Recognizing the spectrum of sex characteristics highlights the imperative for individualized, affirming care and challenges policies or interventions grounded in rigid binary assumptions. Fausto-Sterling [39] emphasizes that, given potential discrepancies arising at different stages of sexual development, any attempt to rigidly and definitively classify a newborn’s sex based solely on genitalia is inadequate. Furthermore, this complexity expands because sexual development does not stop at birth. Adults surrounding the newborn will identify their sex by their genitalia, initiating the process of gender socialization, but the individual’s sexual identity—also called gender identity, defined as a person’s self-perception as male, female, or non-binary [21,38]—may not correspond to that assignment.

Discrepancies between assigned sex and experienced gender can give rise to transgender or non-binary identities, encompassed under the broader term *trans*, which are frequently marginalized by healthcare systems rooted in binary assumptions [5]. Historically, the term “transsexual” referred to individuals seeking medical interventions (such as hormones or surgery) to change their physical characteristics and align their bodies with their gender identity. In contrast, “transgender” is broader, is broader, including all whose gender identity does not align with their assigned sex, regardless of medical transition. Currently, the inclusive term “trans” encompasses transgender, transsexual, and related identities, including non-binary experiences [5,40]. The evolution of language reflects a deeper understanding of this diversity.

Gender identity constitutes a key social determinant of health, determinant of health, shaping access to care, well-being, and life opportunities through interactions with structural inequalities. Healthcare systems rooted in binary assumptions have historically marginalized transgender and non-binary populations, creating both systemic and interpersonal barriers [17,18]. Empirical studies document these inequities: transgender individuals frequently experience misgendering, discrimination, and non-affirming environments that discourage care [17,18]. In Spain, for example, 33% of transgender people avoid healthcare services because their preferred name is not on official records, resulting in misidentification in waiting rooms; 58% report discriminatory treatment by healthcare staff, and nearly half have delayed or canceled appointments for these reasons. Moreover, 33% rarely or never receive care that respects their gender identity, and 75% perceive healthcare professionals as having low to very low knowledge about transgender populations [17]. Similar findings are reported by Mahowald et al. [18], highlighting poorer experiences of transgender patients compared to cisgender patients, often due to inadequate professional training. These data underscore the pervasive impact of binary-based healthcare models and the urgent need for inclusive, affirmative, and well-trained systems that recognize and respond to gender diversity.

These barriers stem from a long history of pathologization. For decades, institutions such as the World Health Organization (WHO) and the American Psychiatric Association (APA) classified trans identities and homosexuality as mental disorders, attributing them to causes such as childhood trauma. Although homosexuality was removed from the DSM-III-R (1987) [41] and ICD-10 (1992) [42], trans identity was not formally depathologized until the publication of the ICD-11 (2019) [43] and DSM-5 (2013) [44], a crucial step toward a rights-based model.

To achieve health equity, it is essential to transition to an affirmative approach that validates identities and promotes self-determination [38,45,46]. This entails not only implementing inclusive policies and laws that guarantee legal recognition of gender identity but also adopting respectful, person-centered clinical practices. Recognizing gender identity as a pillar of public health is the first step to dismantling exclusionary structures and building a truly just and universal healthcare system.

## 3. Education and Professional Training for Health Equity

Science exists to understand the world, sometimes complicating life for those who prefer simpler views. Decades of research have demonstrated that sex is neither a simple nor binary category but encompasses multiple dimensions and related characteristics such as gender identity—how we perceive ourselves—and sexual orientation—towards whom we feel attraction [19,47]. Some psychological frameworks expand this model into five components, also differentiating between physical and emotional attraction, and incorporating gender expression, i.e., the way a person socially communicates their identity [21]. These dimensions are influenced by a complex interaction of chromosomal, genetic, epigenetic, hormonal, biochemical, and environmental factors, whose combination results in a broad diversity of phenotypes within human variability [48,49].

Despite scientific consensus on sex-gender diversity, training in healthcare, social, and legal fields remains deficient. This educational gap directly impacts the quality of care received by transgender, intersex, and non-binary individuals, who continue to face systemic barriers to accessing adequate health services. Surveys reflect this deficiency: less than 30% of psychology professionals feel familiar with gender issues [38], and medical students recognize their preparation to attend to trans patients as insufficient [8]. Indeed, actual knowledge of transgender health, rather than self-perceived knowledge, predicts clinical competence, suggesting that superficial familiarity may hinder rather than facilitate inclusive care efforts [50]. Studies demonstrate that lower comfort working with transgender individuals correlates with higher transphobia, while prior education on transgender health associates with lower transphobia scores [50].

In nursing, recent curriculum reviews [51] have revealed significant shortcomings. Formal curricula often limit content to isolated interventions, such as simulations, which in some cases perpetuate stigma by erroneously associating transgender and non-binary people with mental health problems. Textbook and lecture contents are scarce, outdated, or relegated to marginal notes, depriving students of the necessary tools to provide culturally safe care. Students themselves have expressed concern about the lack of opportunities to learn and practice cultural humility related to transgender people, noting that the absence of well-facilitated content stems not from malice but from faculty’s lack of training and understanding [52]. Furthermore, the hidden curriculum—sustained by institutional norms, policies, and structures—perpetuates invisibility and silence regarding transgender individuals, reinforcing cisnormativity [51,53].

These training deficiencies are not only a professional problem but also a structural one. Evidence shows that health inequities affecting these populations cannot be resolved without the transversal integration of sex-gender diversity in curricula, especially in disciplines linked to health, social intervention, and law [38,54]. This need has long been recognized in specialized fields such as endocrinology [55], and more recently it has begun to be normatively incorporated in contexts such as Spain. Specifically, Law 2/2021 of June 7 on social equality and non-discrimination on grounds of gender identity, gender expression, and sexual characteristics mandates in Article 35 the incorporation of this diversity at all educational levels. Paragraph 3 requires educational materials to promote respect for and protection of the right to gender identity and expression, as well as sexual characteristics. Paragraph 4 mandates that curricula include appropriate pedagogies for recognizing diverse genital configurations and their relationship to identities, integrating transgender and intersex content transversally and specifically [37].

Transforming educational systems through a sex-gender diversity lens is not merely a scientific update but an ethical obligation directly affecting the right to health, equity, and dignity for all people. This transformation implies revising not only curricular content but also pedagogical practices and institutional environments. The following are the main strategies proposed by professional associations to advance more inclusive education:

### 3.1. Review of Foundations and Language

The first step consists of questioning and redefining the conceptual frameworks from which human biology is taught, overcoming the binary reductionism that has historically dominated scientific discourse. It is essential to update academic and clinical language to be inclusive, precise, and respectful. For example, it is recommended to refer to reproductive systems by their functional anatomical terms, such as the Testicular System and the Ovo-Uterine System [1,14], rather than assigning gendered labels that reinforce stereotypes [56].

Although traditionally simplified based on sexual development, labeling organs as “female” or “male” imposes binary which can contribute to the invisibilization oftransgender and some intersex [57,58]. It is therefore essential to provide comprehensive training for educators and healthcare professionals on the correct use of individuals’ names and pronouns, as well as the implementation of neutral, inclusive language that avoids assumptions based on binary sex or gender categories [59]. A notable example of this transformation is the Reproductive Physiology Module developed by ADInstruments on its Lt platform [60,61], which employs expressions like “Ovo-uterine Reproductive Physiology” and “Testicular Reproductive Physiology” in its syllabus, as well as offering a pre-lab distinguishing genotypic sex, phenotypic sex, and gender identity. This resource has been successfully implemented in universities such as Queen’s University Belfast, where it has been integrated into the curriculum with faculty and student participation to promote physiology teaching that reflects diversity and supports gender inclusion.

### 3.2. Curricular Integration

Sex-gender diversity must be integrated transversally at all levels and disciplines of the educational system, not treated as an isolated or complementary topic. This integration should encompass both theoretical content and pedagogical practices, fostering a critical, inclusive, and up-to-date understanding of social and bodily realities. A relevant example is the set of learning outcomes published by the Human Anatomy and Physiology Society (HAPS [62] for introductory university physiology courses. These competency-based outcomes focus on key concepts and skills students should develop, representing an explicit effort to incorporate sex-gender diversity into biomedical education aligned with human rights. Their objectives include defining and differentiating terms such as sex, gender, gender identity, and sexual orientation; describing hormonal therapy effects in transgender people and explaining the “second puberty”; contrasting cisgender and transgender concepts and analyzing implications of assigned sex at birth; and explaining differences in sexual development, among others.

It is recommended to enrich curricula through experiential learning (simulations, case studies, direct interaction with transgender and intersex people) and resources created by the LGTBIQA+ community itself, as well as conducting curricular “mapping” to identify and remedy educational gaps. Additionally, training should actively promote affirmative approaches, such as affirmative therapy, whose benefits for transgender people’s wellbeing are well established [45], and explicitly condemn conversion therapies and other harmful, scientifically unsupported practices [38,63,64].

The informal curriculum, based on everyday interactions, also reflects deficiencies due to faculty’s lack of knowledge and preparation. Emphasis is placed on training teaching staff in inclusive practices, avoiding microaggressions, and creating safe learning environments. Tools such as AQUERY [51] help repair relationships after errors in name or pronoun use in nursing care, fostering a climate of respect and continuous improvement.

This approach marks significant progress in training professionals capable of understanding bodily, hormonal, and identity diversity from a scientific and ethical perspective. To ensure the relevance and legitimacy of these contents, it is essential to include the voices of transgender and intersex people in curricular design [54], ensuring education that not only informs but also advances community health, equity, and human rights.

### 3.3. Competency Development and Creation of Safe Spaces

The development of competencies in diversity must be structurally and transversally integrated into curricula to ensure that students acquire the necessary skills to provide culturally safe, respectful care aligned with the plural realities of contemporary societies. This entails not only training in cultural competence and promoting critical analysis of personal biases, but also a thorough review of academic programs, materials, and documents to identify and eliminate cisnormative content that perpetuates exclusionary models [56]. Such an approach is essential for future professionals to understand the intersectionality of identities and to act ethically, effectively, and committedly toward human diversity [19,38].

Within this framework, creating inclusive environments represents a core component of institutional commitment to equity. A particularly relevant strategy to guarantee this is the design and implementation of gender-neutral restrooms. Evidence indicates that the availability of such spaces significantly contributes to the psychological well-being of transgender and non-binary individuals by fostering greater perceptions of safety, dignity, and recognition [65]. By reducing anxiety, dysphoria, and anticipated stigma, gender-neutral restrooms not only promote a more affirming experience of the environment but also strengthen a sense of belonging and social cohesion in educational and workplace settings, with positive impacts on productivity and academic performance [65]. Moreover, these spaces benefit other groups, such as persons with disabilities requiring assistance from caregivers of a different gender, or racialized individuals for whom inclusive facilities reduce experiences of discrimination and stigmatization [66]. Thus, the adoption of gender-neutral restrooms transcends practical value to become a symbol of institutional commitment and a structural measure to overcome the gender binary embedded in traditional architecture. In Spain, regulations such as Law 4/2023 [67] and Royal Decree 1026/2024 [68] urge institutions to assess and adapt their facilities within the framework of LGTBIQA plans to eliminate both physical and symbolic barriers. Conversely, the absence of inclusive spaces perpetuates structural exclusion, reinforces binary norms, and associates with negative physical and mental health outcomes—including urinary tract infections, anxiety, depression, and increased risk of suicidal behaviors—particularly among transgender and non-binary individuals who avoid these spaces or face harassment when using them [69].

Creating inclusive environments must extend beyond physical spaces to encompass administrative and management systems. Adapting forms, records, and databases to enable self-identification through open-text options, as well as the use of chosen names and preferred pronouns, constitutes a fundamental practice to reflect and respect gender diversity [38,56]. In this regard, including pronouns in attendance lists or academic forms is emerging as a best practice within academic institutions and professional platforms such as LinkedIn, contributing to the normalization of respect for personal identity across contexts.

Furthermore, advancing toward truly inclusive education requires updating institutional policies and explicitly incorporating these practices into accreditation standards. Institutions bear the responsibility to promote ongoing faculty training in communication skills grounded in respect for names, pronouns, and identities, while fostering genuine and diverse representation within faculty and student bodies [51]. These actions not only reflect a commitment to human rights and equity but also decisively contribute to creating educational spaces that are safe, accessible, and respectful for all individuals, regardless of gender identity or expression.

## 4. Key Areas for Healthcare

Affirmative healthcare is a comprehensive approach aimed at respecting, supporting, and validating each person’s gender identity throughout all aspects of their care, contrasting with historical practices that pathologized diversity [38]. The success of this model depends on healthcare professionals acting based on robust evidence and genuine clinical competence, rather than superficial familiarity [50], overcoming biases and prejudices to provide quality care.

Within this framework, medical interventions such as Gender-Affirming Hormone Therapy (GAHT) and Gender-Affirming Surgery (GAS) are fundamental pillars. GAHT, involving the administration of estrogens and antiandrogens for trans women or testosterone for trans men, facilitates the alignment of secondary sexual characteristics with gender identity [1,2]. Concurrently, gender-affirming surgeries enable anatomical congruence. Both interventions have demonstrated highly positive impacts not only on physical health but also on mental health, with significant improvements in quality of life and notable reductions in anxiety and depressive symptoms [70,71].

Psychological support constitutes the other cornerstone of affirmative care, focusing on validating identity and equipping individuals with tools to manage distress stemming from social stigma. Numerous studies indicate that identity validation through affirmative therapy is effective and produces positive outcomes for mental health and well-being among LGTBIQAindividuals, reducing anxiety, depression, internalized transphobia, and experiences of non-affirmation [46]. In contrast, conversion therapy comprises a set of pseudoscientific treatments aimed at changing either a homosexual or bisexual person’s sexual orientation to heterosexual or modifying one’s gender identity to match the sex assigned at birth [8]. However, evidence shows that gender identity cannot be altered through psychological or psychiatric therapies [55]. This practice has been widely discredited by medical and psychological organizations, including the American Psychiatric Association (APA), due to its lack of scientific validity and potential to cause severe harm, such as loneliness, substance abuse, depression, anxiety, lack of support, suicidal ideation, and suicide attempts [72,73,74]. Despite its rejection by professional bodies and legal prohibition in several countries—such as Spain—where it is considered unethical, harmful, and empirically invalid, conversion therapy continues to receive some clinical and academic support [63,64]. In this context, it is critical to debunk myths like “massive detransition,” often propagated by conservative sectors without scientific foundation or validity [38,64]. Scientific evidence indicates that detransition is rare (~1%) and often driven by familial pressure, lack of social support [70], or the discovery of a non-binary identity rather than an attempt at “cure” [75]. APA guidelines advocate for a cautious approach that supports exploration of identity rather than premature transition [38,72]. Experts warn that the harm caused by preventing transition is often more irreversible than that of a possible detransition [76].

Beyond these core interventions, a holistic health approach must address specific disparities frequently overlooked. For instance, transgender individuals exhibit a higher prevalence of oral health issues, influenced not only by hormonal factors but also by structural barriers and discrimination leading to avoidance of dental care [77]. Similarly, they face a significantly increased risk of developing eating disorders compared to cisgender populations, necessitating specialized attention [78]. It is also essential to counteract social prejudices with scientific data, such as in the debate on athletic performance, where current evidence does not support claims of inherent competitive advantage in trans women undergoing standard hormone therapy [79,80].

To operationalize this comprehensive care, healthcare settings must implement practical and structural measures. These include mandatory and ongoing training for personnel, alongside the implementation of registration systems that allow patients to use their chosen name and gender identity across all forms and prescriptions. Respectful communication, including correct inquiry and use of pronouns, is an essential clinical practice. Furthermore, creating physically safe environments—such as gender-neutral restrooms—and adapting services—for example, providing gynecological exams outside exclusively female spaces—are fundamental to ensuring equitable and dignified access for all individuals [81].

## 5. Research and Indicators for Inclusive Health

Inclusive research is a fundamental approach aimed at identifying and making visible the inequalities faced by LGTBIQA+ individuals, in order to design fairer and more equitable public policies [19]. Sex, gender identity, and sexual orientation are central to a person’s self-understanding and shape their experiences and opportunities. These demographic characteristics structure people’s lives and may create gender-based inequalities manifested in forms such as segregation, discrimination, violence, sexism, homophobia, and transphobia. Understanding their effects is crucial; however, this requires the use of valid measures that encompass the full spectrum of possibilities.

Data collection frequently assumes these constructs to be simple, binary, and mutually determined at birth, obscuring a more complex reality. Such assumptions canchallenge respondents and include invalidating or offensive terminology. These measurement limitations are not merely academic; they have tangible consequences for sexual and gender minorities in healthcare and social services [82]. Additionally, healthcare professionals’ personal beliefs—including implicit and explicit biases- directly influence the quality of care. Transphobia, cisnormative assumptions, or limited knowledge of gender diversity can result in discriminatory practices [54,83,84]. These dynamics underscore the necessity for ongoing self-reflection, recognition of cisgender privilege, cultural humility, and training in inclusive, rights-based practices, both in clinical and research settings, to avoid reproducing stereotypes or pathologizing identities [38,83]. Futhermore, studies on inclusive practices highlight the critical importance of positioning oneself and examining norms and power structures at the individual, interpersonal, and systemic levels, including heath education, white supremacy, and the cisheteropatriarchy [56].

A gender-diverse perspective challenges the authority of “expert witnesses” to determine others’ identities and upholds self-identification as the sole legitimate means of personal identification [85]. Adopting this perspective aligns with a human rights framework, as articulated in the Yogyakarta Principles, which provide ethical, anti-discriminatory guidelines for data collection to promote social justice [86].

Achieving these objectives necessitates rigorous methodological strategies. The most widely endorsed is the two-step method, recommended by organizations such as the World Professional Association for Transgender Health [87] and the U.S. National Academies [82,88]. This approach separately inquires about sex assigned at birth (“What sex were you assigned at birth?”) and current gender identity (“How do you describe yourself?”). It is advisable to reconsider which aspects best explain the phenomenon under study, including as specific questions as possible and offering non-binary options (see Table 1 for a two-step method example). Terms such as “natal sex” or “birth sex” are considered pejorative, as noted in the American Psychiatric Association style guide [89]. The same guide prefers using “gender” alone when referring to social groups. To improve question development, consulting the National Academies report *Measuring Sex, Gender Identity, and Sexual Orientation* [82] and the practical guide *Inclusion of Sexual and Gender Diversity Perspectives in Research* [19] is recommended. These documents emphasize the need for validated measures and propose areas for future research.

Inclusive research is challenging paradigms notably in areas like sexual dimorphism in neuroscience. Traditionally, differences in the prevalence of neurobehavioral disorders were attributed to inherent brain differences. Although differences exist in total brain size, meta-analyses confirm that structural discrepancies are minimal when controlling for body size [48]. The notion of a “male brain” versus a “female brain” appears more a product of theoretical models than robust neuroanatomical reality. For instance, studies in pain research show that transgender women’s response patterns resemble cisgender women’s more than men’s, underscoring the influence of gender identity on physiological responses [15,90]. These findings demand the inclusion of gender identity as a variable to understand the complex interplay of genetic, epigenetic, and experiential factors.

Conducting such research effectively requires not only validated instruments but also dedicated funding programs ([57] and rigorous training for research teams [82]. on the sensitivity and competence of those collecting and analyzing information. Results should be disseminated transparently and accessibly, with active engagement of LGTBIQA+communities ensuring research fosters equitable science and informs inclusive public health policies.

## 6. Conclusions: Health with an Inclusive Sex-Gender Approach

This Perspective set out to critically examine the limitations of the binary biomedical framework and to explore pathways for embedding sex-gender diversity into healthcare. The evidence reviewed confirmsthat the traditional biomedical model—based on a binary understanding of sex and gender—may be insufficient to address human diversity and contributes to significant health inequities affecting transgender, intersex, and non-binary individuals. Historical pathologization, gaps in professional training, and exclusionary research practices have created to systemic barriers to equitable and respectful healthcare. Addressing these challenges requires integrating sex-gender diversity as a central element across health policies and practices.

Meaningful transformation must occur simultaneously across multiple domains. In healthcare and public policy, pathologizing frameworks should be abandoned in favor of evidence-based, affirmative care that respects self-determination. In education, structural reforms of health and social sciences curricula are needed to equip professionals with inclusive knowledge, cultural humility, and appropriate language to serve diverse populations. In research, inclusive methodologies and metrics—such as the two-step method—should be adopted, with teams trained to generate valid data that illuminate health inequities and inform effective policy development.

Future efforts should prioritize longitudinal and intersectional studies to capture the evolving needs of gender-diverse populations across the lifespan. Development of culturally sensitive measurement tools will enhance data accuracy and inclusivity. Interprofessional training programs, co-designed with transgender and intersex communities, will foster clinical environments that are competent, and affirming. Policy frameworks should remain adaptable and responsive to emerging evidenceto dismantle structural barriers and promote equity.

Advancing this inclusive paradigm is not only a response to historical inequities but also essential for building rigorous scientific knowledge, effective healthcare systems, and socially inclusive environments for all individuals.

## Figures and Tables

**Table 1 healthcare-13-02440-t001:** Example of the Two-Step Method.

Item	Options
What sex were you assigned at birth?	Female, Male, Intersex
How do you identify yourself?	Man, Woman, Non-binary/Other

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
