# Peer review of "Beyond Gender Binarism: Implications of Sex-Gender Diversity for Health Equity"

_healthcare, 2025, doi:10.3390/healthcare13192440_

Round 1

Reviewer 1 Report

Comments and Suggestions for Authors

The paper aims to review how the traditional binary model of sex and gender in healthcare contributes to health inequities for transgender and intersex individuals. While the topic is important and timely, the manuscript needs significant revisions before it can be considered for publication.

Although the paper identifies itself as a review, it does not follow the standard structure or methodological clarity expected of a review article. The abstract mentions a "theoretical review," but there is no explanation of how the literature was selected or analyzed.

To strengthen the paper and ensure it meets the expectations for a review, the authors should 1) clearly describe the review methodology, 2) explain how sources were selected, 3) define the type of review conducted, and 4) Provide inclusion and exclusion criteria. 

Without this information, the manuscript reads more like an opinion or perspective piece rather than a formal review.

Comments on the Quality of English Language

The paper aims to review how the traditional binary model of sex and gender in healthcare contributes to health inequities for transgender and intersex individuals. While the topic is important and timely, the manuscript needs significant revisions before it can be considered for publication.

Although the paper identifies itself as a review, it does not follow the standard structure or methodological clarity expected of a review article. The abstract mentions a "theoretical review," but there is no explanation of how the literature was selected or analyzed.

To strengthen the paper and ensure it meets the expectations for a review, the authors should 1) clearly describe the review methodology, 2) explain how sources were selected, 3) define the type of review conducted, and 4) Provide inclusion and exclusion criteria. 

Without this information, the manuscript reads more like an opinion or perspective piece rather than a formal review.

Author Response

Comments 1: 

The paper aims to review how the traditional binary model of sex and gender in healthcare contributes to health inequities for transgender and intersex individuals. While the topic is important and timely, the manuscript needs significant revisions before it can be considered for publication.

Although the paper identifies itself as a review, it does not follow the standard structure or methodological clarity expected of a review article. The abstract mentions a "theoretical review," but there is no explanation of how the literature was selected or analyzed.

To strengthen the paper and ensure it meets the expectations for a review, the authors should 1) clearly describe the review methodology, 2) explain how sources were selected, 3) define the type of review conducted, and 4) Provide inclusion and exclusion criteria. 

Without this information, the manuscript reads more like an opinion or perspective piece rather than a formal review.

Response: 

I sincerely thank the reviewers for their constructive comments. I have carefully considered the feedback regarding both the abstract and the nature of the manuscript.

  1. Regarding the abstract. I have revised the abstract to align with the Perspective format, avoiding an IMRaD structure. The new version improves sentence structure, precision of language, and academic tone. It now provides readers with a clear overview of the manuscript, focusing on recent developments, the limitations of the binary biomedical model, and proposed directions for systemic transformation.
  2. Regarding the nature of the manuscript. I acknowledge the reviewers’ observation that the original version did not include a systematic search strategy or selection criteria for sources. Therefore, I have reconceptualized the manuscript as a Perspective article, which is consistent with the scope of the journal. In this revised version, I situate the critical reflection within the most recent literature, highlight advances in medical education, affirmative care, and inclusive research, and propose concrete directions for future health system transformation.

I believe these changes strengthen the clarity, academic rigor, and coherence between the abstract, the manuscript’s structure, and the objectives of the special issue. I have also discussed the change of article type with the editor and implemented revisions guided by the comments of other reviewers, which update and strengthen the theoretical framework and include a general review of the technical English writing style.

Reviewer 2 Report

Comments and Suggestions for Authors

I believe that the abstract follows an appropriate structure (introduction, methods, results, and conclusions), which facilitates overall understanding of the study. The topic is relevant and current, and the objective of the review is clearly stated. In terms of writing, I note that some sentences are long and would benefit from clearer punctuation or restructuring to improve readability. For example, the sentence: ‘This review aims to analyse the limitations of this model and synthesise the current evidence to propose a comprehensive approach that promotes equity in health service delivery, professional training and research’ could be divided or simplified to avoid ambiguity. I therefore believe that improving the structure of the sentences, the precision of the language and the academic tone would strengthen its clarity and rigour.

As for the introduction, I believe it presents a relevant and up-to-date overview of the evolution of the understanding of the concepts of sex and gender, highlighting the shift from a binary and essentialist approach to a more complex, multidimensional and spectral perspective. I find it appropriate that the introduction contextualises the topic from a historical and conceptual perspective, correctly differentiating between sex (as a biological construct) and gender (as a social and cultural construct), and drawing on specialised literature.

In terms of content, I consider the theoretical framework to be well-founded, especially in pointing out how the binary model persists in social and institutional contexts, including health and education systems. The problem is addressed not only from a biological perspective, but also from a human rights and public health perspective, which gives the text depth and interdisciplinary relevance. The inclusion of concepts such as ‘cissexism,’ ‘medical pathologisation,’ and ‘epistemological biases’ enriches the discussion and demonstrates a critical stance towards traditional biomedical practices.

However, I believe that the text could be improved with more precise and concise wording in some parts, as I find some sentences to be excessively long, which hinders fluent reading and weakens the argumentative force of certain key ideas. For example, expressions such as ‘the consequences are not only harmful to transgender, non-binary and intersex people, but also to those who do not fully conform to hegemonic gender norms...’ could be rephrased to improve clarity and avoid repetition.

Furthermore, although the concepts addressed are relevant, the progression between paragraphs could be more cohesive, especially in the transition from the historical discussion on sex/gender to the practical implications in the field of health. I believe it would be useful to incorporate clearer linking sentences to guide the reader and reinforce the connection between the theoretical framework and the issue of health inequalities.

I also recommend avoiding certain expressions that, although rhetorically powerful, require greater conceptual precision to suit the academic tone of a scientific publication. For example, the phrase ‘symbolically violent forms of invisibilisation’ could be reworded using more precise terminology and, if possible, accompanied by a citation or reference to support its inclusion.In summary, the introduction offers a critical, well-documented and conceptually sound overview. However, it could be strengthened through more concise writing, greater fluidity of argument between sections, and more rigorous use of academic language.

In the methods section, although this is not an empirical study, the theoretical and documentary review approach is adequately described. The article indicates the thematic areas reviewed (sexual development, social determinants, affirmative care, vocational training, and inclusive research) and mentions the use of guidelines and regulatory frameworks as part of the analysis. However, I believe it would be useful to specify more precisely the criteria for including sources, the review period, or whether systematic or selective strategies were applied to ensure the methodological rigour of the review process.

With regard to the conclusions section, I believe that it clearly and coherently summarises the main findings and arguments developed throughout the article. It adequately reiterates the central idea that the traditional biomedical model, by maintaining a binary conception of sex and gender, is insufficient to address the complexity of human diversity and contributes to the perpetuation of structural inequalities in health. The conclusion integrates the different dimensions analysed (clinical practice, professional training, public policies and research) and proposes a systemic transformation with a cross-cutting approach to sex and gender diversity.

In terms of content, I appreciate the identification of concrete actions for each area: abandoning pathologising frameworks and adopting an affirmative approach in healthcare; profound curricular reform in the health and social sciences; and the adoption of inclusive methodologies in research, such as the two-step method. I also welcome the proposal for a future agenda focused on longitudinal and intersectional studies, as well as the development of culturally sensitive measurement tools and interprofessional training strategies that actively involve trans and intersex communities.

However, in terms of style and wording, I believe that some paragraphs could be improved with a more concise structure and more direct language. For example, the sentence ‘this transformation calls for continued investment in longitudinal and intersectional studies...’ is lengthy and could be broken down to facilitate understanding. In addition, certain expressions such as ‘moving towards this inclusive paradigm is not merely a matter of historical redress...’ could be rephrased in a more objective and technical tone, in keeping with the academic style.

I also suggest avoiding unnecessary repetition of certain terms or structures (for example, ‘competent, respectful, and affirmative’ appears frequently and could be varied slightly in its wording), and ensuring that terminological consistency is maintained throughout the section (for example, consistently using ‘people with diverse gender identities’ rather than alternating multiple labels without clarification).

Overall, I believe that this article makes a valuable, critical and timely contribution to the debate on the inclusion of gender diversity in the field of health. It provides a well-founded review, with a multidisciplinary approach and concrete proposals for transforming entrenched institutional practices. To strengthen its academic impact and facilitate understanding, I recommend improving the structural coherence between sections, conceptual precision and conciseness in the writing. I believe that these adjustments would optimise the scientific rigour and broaden the scope of the article within the academic and professional community.

Author Response

Comment 1: I believe that the abstract follows an appropriate structure (introduction, methods, results, and conclusions), which facilitates overall understanding of the study. The topic is relevant and current, and the objective of the review is clearly stated. In terms of writing, I note that some sentences are long and would benefit from clearer punctuation or restructuring to improve readability. For example, the sentence: ‘This review aims to analyse the limitations of this model and synthesise the current evidence to propose a comprehensive approach that promotes equity in health service delivery, professional training and research’ could be divided or simplified to avoid ambiguity. I therefore believe that improving the structure of the sentences, the precision of the language and the academic tone would strengthen its clarity and rigour.

Comment 2: As for the introduction, I believe it presents a relevant and up-to-date overview of the evolution of the understanding of the concepts of sex and gender, highlighting the shift from a binary and essentialist approach to a more complex, multidimensional and spectral perspective. I find it appropriate that the introduction contextualises the topic from a historical and conceptual perspective, correctly differentiating between sex (as a biological construct) and gender (as a social and cultural construct), and drawing on specialised literature.

Response 2: I have updated the manuscript to clarify the concepts of sex and gender based on recent and relevant literature.

Comment 3: In terms of content, I consider the theoretical framework to be well-founded, especially in pointing out how the binary model persists in social and institutional contexts, including health and education systems. The problem is addressed not only from a biological perspective, but also from a human rights and public health perspective, which gives the text depth and interdisciplinary relevance. The inclusion of concepts such as ‘cissexism,’ ‘medical pathologisation,’ and ‘epistemological biases’ enriches the discussion and demonstrates a critical stance towards traditional biomedical practices.

Response 3: I have updated several references and introduced new ones in the introduction, guided by the comments of other reviewers, which further strengthen and support your valuable insights

Comment 4: However, I believe that the text could be improved with more precise and concise wording in some parts, as I find some sentences to be excessively long, which hinders fluent reading and weakens the argumentative force of certain key ideas. For example, expressions such as ‘the consequences are not only harmful to transgender, non-binary and intersex people, but also to those who do not fully conform to hegemonic gender norms...’ could be rephrased to improve clarity and avoid repetition.

Furthermore, although the concepts addressed are relevant, the progression between paragraphs could be more cohesive, especially in the transition from the historical discussion on sex/gender to the practical implications in the field of health. I believe it would be useful to incorporate clearer linking sentences to guide the reader and reinforce the connection between the theoretical framework and the issue of health inequalities.

Response 4: I have thoroughly revised the manuscript’s writing style, improving clarity, conciseness, and sentence structure throughout. Long or repetitive sentences, including the example you highlighted, have been rephrased to enhance readability and strengthen the argumentative impact of key ideas. I have especially revised Sections 1 and 2 to improve the flow and coherence, ensuring a clear connection between the historical discussion of sex and gender and the subsequent examination of health inequities.

Comment 5: I also recommend avoiding certain expressions that, although rhetorically powerful, require greater conceptual precision to suit the academic tone of a scientific publication. For example, the phrase ‘symbolically violent forms of invisibilisation’ could be reworded using more precise terminology and, if possible, accompanied by a citation or reference to support its inclusion.In summary, the introduction offers a critical, well-documented and conceptually sound overview. However, it could be strengthened through more concise writing, greater fluidity of argument between sections, and more rigorous use of academic language.

Response 5: I have revised the sentence as suggested and strengthened the argument with additional references. The updated text now reads:

"Although traditionally simplified based on sexual development, labeling organs as “female” or “male” imposes binary gender assumptions on organs that have none and can be present in both men and women, which can contribute to the invisibilization of transgender and some intersex individuals [53,54]. It is therefore essential to provide comprehensive training for educators and healthcare professionals on the correct use of individuals’ names and pronouns, as well as the implementation of neutral, inclusive language that avoids assumptions based on binary sex or gender categories [55]."

Comment 6: In the methods section, although this is not an empirical study, the theoretical and documentary review approach is adequately described. The article indicates the thematic areas reviewed (sexual development, social determinants, affirmative care, vocational training, and inclusive research) and mentions the use of guidelines and regulatory frameworks as part of the analysis. However, I believe it would be useful to specify more precisely the criteria for including sources, the review period, or whether systematic or selective strategies were applied to ensure the methodological rigour of the review process.

Response 6: I clarify that the review focused on recent references across three key areas related to healthcare competence: professional training, clinical practice, and research. I have carefully considered the feedback of other reviewers regarding both the abstract and the nature of the manuscript:

Nature of the manuscript: The manuscript has been reconceptualized as a Perspective article, consistent with the journal’s scope. The revised version situates the critical reflection within recent literature, highlighting advances in medical education, affirmative care, and inclusive research, and proposes concrete directions for future health system transformation.

Abstract: I revised it to align with the Perspective format. The new version improves sentence structure, precision, and academic tone, offering readers a clear overview of recent developments, the limitations of the binary biomedical model, and proposed directions for systemic transformation.

I have also discussed the change of article type with the editor and implemented revisions guided by the comments of other reviewers.

Comment 7: With regard to the conclusions section, I believe that it clearly and coherently summarises the main findings and arguments developed throughout the article. It adequately reiterates the central idea that the traditional biomedical model, by maintaining a binary conception of sex and gender, is insufficient to address the complexity of human diversity and contributes to the perpetuation of structural inequalities in health. The conclusion integrates the different dimensions analysed (clinical practice, professional training, public policies and research) and proposes a systemic transformation with a cross-cutting approach to sex and gender diversity.

In terms of content, I appreciate the identification of concrete actions for each area: abandoning pathologising frameworks and adopting an affirmative approach in healthcare; profound curricular reform in the health and social sciences; and the adoption of inclusive methodologies in research, such as the two-step method. I also welcome the proposal for a future agenda focused on longitudinal and intersectional studies, as well as the development of culturally sensitive measurement tools and interprofessional training strategies that actively involve trans and intersex communities.

However, in terms of style and wording, I believe that some paragraphs could be improved with a more concise structure and more direct language. For example, the sentence ‘this transformation calls for continued investment in longitudinal and intersectional studies...’ is lengthy and could be broken down to facilitate understanding. In addition, certain expressions such as ‘moving towards this inclusive paradigm is not merely a matter of historical redress...’ could be rephrased in a more objective and technical tone, in keeping with the academic style.

I also suggest avoiding unnecessary repetition of certain terms or structures (for example, ‘competent, respectful, and affirmative’ appears frequently and could be varied slightly in its wording), and ensuring that terminological consistency is maintained throughout the section (for example, consistently using ‘people with diverse gender identities’ rather than alternating multiple labels without clarification).

Overall, I believe that this article makes a valuable, critical and timely contribution to the debate on the inclusion of gender diversity in the field of health. It provides a well-founded review, with a multidisciplinary approach and concrete proposals for transforming entrenched institutional practices. To strengthen its academic impact and facilitate understanding, I recommend improving the structural coherence between sections, conceptual precision and conciseness in the writing. I believe that these adjustments would optimise the scientific rigour and broaden the scope of the article within the academic and professional community.

Response 7: Thank you for your detailed and constructive feedback on the conclusions section. I have carefully revised this section to address your observations regarding style, conciseness, and terminology.

Specifically, I have:

  • Streamlined lengthy sentences to enhance readability and facilitate comprehension. For example, the previous sentence on future research initiatives has been divided and clarified for directness.
  • Adjusted expressions to adopt a more technical and objective tone, replacing phrases such as “moving towards this inclusive paradigm is not merely a matter of historical redress” with language emphasizing rigor, evidence-based practice, and systemic transformation.
  • Reduced repetitive phrasing, such as “competent, respectful, and affirmative,” and ensured consistent use of the term “people with diverse gender identities” throughout the section.
  • I have carefully revised the manuscript to improve structural coherence between sections, enhance conceptual precision, and increase conciseness throughout the text.

Reviewer 3 Report

Comments and Suggestions for Authors

Dear author,

I enjoyed reading the manuscript, and I congratulate you on the importance of the topic you chose. With a view to improvement, I present below a set of notes:

Title: is clear and concise and reflects the main focus of the study.

Abstract: has a different structure from the one used in the manuscript, creating the impression that the IMRAD structure was used. Please modify the abstract so that readers can get an overall idea of the content and structure.

Keywords: well chosen.

Introduction

Lines 104-113: This paragraph does not cite any sources of information. Please include them.

Lines 170-175: Please add citations to these statements.

Lines 212-215: Please provide citations to support this statement.

Line 276: Please include the citation for “ADInstruments on its Lt platform” so readers can explore this content.

Conclusions

What objective or question do these conclusions address? Please clarify.

Please review the methodology used in your paper, considering that, according to Machi and McEvoy (2022), the theoretical review “seeks to identify existing theories and interrelationships among theories addressing a pursuit phenomenon. Its purpose is to create a more refined theory set and develop new hypotheses to enhance and extend prevailing theory. These reviews are usually undertaken when new questions emerge about prior research and stimulate the need for a refinement and a new theoretical understanding. Theoretical reviews are also used to develop a theoretical framework for large accumulations of descriptive and explanatory research studies lacking a theoretical frame.”

Machi, L. A., & McEvoy, B. T. (2022). The Literature Review: Six Steps to Success (4th ed.). Corwin Press.

Overall appreciation: In this manuscript, the author discusses the topic under analysis throughout the subsections, rather than including a section solely dedicated to this discussion. It presents an introduction divided into two sections with a brief contextualization and without presenting the issue to be analysed or the objective. Overall, the manuscript resembles an essay more than a review because the arguments are discussed throughout the manuscript. According to the publisher MDPI, an essay is “an article type commonly used in humanities and social sciences to present provocative arguments aimed at stimulating readers to rethink certain issues. The structure is similar to that of a review. Arguments should be supported by relevant references" (https://www.mdpi.com/about/article_types), as is the case with the American Psychiatric Association (APA). I therefore suggest that the author review the classification assigned to the theoretical review manuscript. Regardless, the introduction of the manuscript would benefit from a more comprehensive context, followed by the presentation of the thesis statement, aim, or the question to be analysed.

Good luck with your research!

Author Response

Comment 1: Abstract: has a different structure from the one used in the manuscript, creating the impression that the IMRAD structure was used. Please modify the abstract so that readers can get an overall idea of the content and structure.

Response 1: Thank you very much for your kind words and for recognizing the relevance of the topic. I truly appreciate your thoughtful reading and feedback. I have revised the abstract to align with the Perspective format, avoiding an IMRaD structure. The new version improves sentence structure, precision of language, and academic tone. It now provides readers with a clear overview of the manuscript, focusing on recent developments, the limitations of the binary biomedical model, and proposed directions for systemic transformation.

Comment 2:  Lines 104-113: This paragraph does not cite any sources of information. Please include them.

Response 2: I have added recent references. Lines 160-164

Comment 3: Lines 170-175: Please add citations to these statements.

Response 3: I have incorporated the recent WHO (2024) reference. Lines 233-249

Comment 4: Lines 212-215: Please provide citations to support this statement.

Response 4: I have added recent references (2025). Line 303

Comment 5: Line 276: Please include the citation for “ADInstruments on its Lt platform” so readers can explore this content.

Response 5: Added in line 367

Comment 6: Conclusions

What objective or question do these conclusions address? Please clarify.

Response 6: I have revised and strengthened the conclusion to ensure clearer alignment with the objectives stated in the abstract. In particular, I have added the following introductory sentence to explicitly frame the purpose of the conclusions: “This Perspective set out to critically examine the limitations of the binary biomedical framework and to explore pathways for embedding sex-gender diversity into healthcare.”

Comment 7: Overall, the manuscript resembles an essay more than a review because the arguments are discussed throughout the manuscript. According to the publisher MDPI, an essay is “an article type commonly used in humanities and social sciences to present provocative arguments aimed at stimulating readers to rethink certain issues. The structure is similar to that of a review. Arguments should be supported by relevant references" (https://www.mdpi.com/about/article_types), as is the case with the American Psychiatric Association (APA). I therefore suggest that the author review the classification assigned to the theoretical review manuscript. Regardless, the introduction of the manuscript would benefit from a more comprehensive context, followed by the presentation of the thesis statement, aim, or the question to be analysed.

Response 7: Sincerely thank the reviewer for the detailed feedback. In response, I have reconceptualized the manuscript as a Perspective article, which aligns with the scope of the journal. In this revised version, I situate the critical reflection within the most recent literature, emphasize advances in medical education, affirmative care, and inclusive research, and propose concrete directions for the future transformation of health systems.

Reviewer 4 Report

Comments and Suggestions for Authors

I would like to applaud this author on a very well written and very well researched review of this crucially important topic. I provider the following brief comments for consideration by the author to further strengthen this already very strong manuscript. 

*Be consistent with the use of LGBT vs LGBTQ vs LGBTI, etc. Select whichever form of the LGBTQ+ acronym that you would like to utilize for this manuscript, and be consistent throughout the document. 

*An area that the author may wish to briefly discuss in this review is the link between colonialism and the gender binary. If they so wish, I would encourage the author to consider the following articles and potentially incorporate these related concepts into their review to demonstrate how western colonialism continues to reinforce the notion of rigid gender binaries, which are at odds not only with modern science, but with non-Western native and indigenous cultures as well:

Wagner, T. L., Marsh, D., & Curliss, L. (2025). Theories and implications for centering Indigenous and queer embodiment within sociotechnical systems. Journal of the American Society for Information Science and Technology, 76(2), 397–412. https://doi.org/10.1002/asi.24746

O’Sullivan, S. (2021). The Colonial Project of Gender (and Everything Else). Genealogy, 5(3), 67. https://doi.org/10.3390/genealogy5030067

*Starting at Line 421 - I also encourage the author to consider a brief discussion on how the personal beliefs of healthcare professionals towards sex and gender diversity often influence how they interact with transgender, gender non-binary, gender-non-conforming, and genderqueer patients and community members, including their political and religious beliefs. Often there remains the notion that healthcare professionals put their personal beliefs and attitudes aside when working with patients, but that is certainly not the case. Therefore, I encourage the author to see if it would be possible to include a brief discussion about this topic in their manuscript, particularly in relation to how healthcare professionals need to engage in personal reflection to identity and understand their own implicit and unconscious biases 

Author Response

Comment 1: Be consistent with the use of LGBT vs LGBTQ vs LGBTI, etc. Select whichever form of the LGBTQ+ acronym that you would like to utilize for this manuscript, and be consistent throughout the document. 

Response 1: Thank you very much for your kind words and constructive feedback. I have addressed your suggestion by standardizing the acronym throughout the manuscript, consistently using LGTBIQA+ in all instances.

Comment 2: An area that the author may wish to briefly discuss in this review is the link between colonialism and the gender binary. If they so wish, I would encourage the author to consider the following articles and potentially incorporate these related concepts into their review to demonstrate how western colonialism continues to reinforce the notion of rigid gender binaries, which are at odds not only with modern science, but with non-Western native and indigenous cultures as well

Response 2: Thank you for providing these valuable references. I have briefly incorporated them into the manuscript to highlight how Western colonialism has reinforced rigid gender binaries, affecting Indigenous and non-Western societies.

Comment 3: Starting at Line 421 - I also encourage the author to consider a brief discussion on how the personal beliefs of healthcare professionals towards sex and gender diversity often influence how they interact with transgender, gender non-binary, gender-non-conforming, and genderqueer patients and community members, including their political and religious beliefs. Often there remains the notion that healthcare professionals put their personal beliefs and attitudes aside when working with patients, but that is certainly not the case. Therefore, I encourage the author to see if it would be possible to include a brief discussion about this topic in their manuscript, particularly in relation to how healthcare professionals need to engage in personal reflection to identity and understand their own implicit and unconscious biases.

Response 3: I have added a brief discussion on this topic in Section 5, Research and Indicators for Inclusive Health, highlighting how healthcare professionals’ personal beliefs and implicit biases can affect care for transgender and gender-diverse individuals, and emphasizing the importance of self-reflection, cultural humility, and inclusive training.

Round 2

Reviewer 1 Report

Comments and Suggestions for Authors

The author has appropriately addressed my comments and suggestions through satisfactory revisions.

Author Response

Dear Reviewer,

I would like to express my sincere gratitude for your careful reading of my manuscript and for the constructive comments you provided. Your observations have been extremely valuable and have significantly contributed to improving the clarity and quality of the paper.

I truly appreciate the time and expertise you dedicated to this review.

With kind regards,
Peter de Jesús

Reviewer 3 Report

Comments and Suggestions for Authors

Dear author,

I really enjoyed rereading your manuscript. The decision to publish a perspective article seems very appropriate to me. I am glad that I was able to contribute to the improvement of your article. I would just suggest that you revise the sentence in lines 547-551, as it seems to be missing a verb. Otherwise, I wish you the best of luck with your research!

Author Response

Dear Reviewer,

Thank you very much for your thoughtful and encouraging feedback. I truly appreciate your kind words and the time you have dedicated to reviewing my manuscript once again. I am glad that the decision to publish it as a perspective article seems appropriate to you, and I am grateful for your valuable contribution to its improvement.

Regarding your suggestion about the sentence in lines 547–551, I carefully reviewed the entire manuscript and have now been able to revise the sentence accordingly and resubmit the manuscript.

Once again, thank you very much for your support and best wishes.

With kind regards,
Peter de Jesús